# The Quality of Counselling for Oral Emergency Contraceptive Pills—A Simulated Patient Study in German Community Pharmacies

**DOI:** 10.3390/ijerph17186720

**Published:** 2020-09-15

**Authors:** Bernhard Langer, Sophia Grimm, Gwenda Lungfiel, Franca Mandlmeier, Vanessa Wenig

**Affiliations:** Department of Health, Nursing, Management, University of Applied Sciences Neubrandenburg, 17033 Neubrandenburg, Germany; sg2807@t-online.de (S.G.); Gwenda.lungfiel@gmx.de (G.L.); franca_m@web.de (F.M.); vanessawenig@gmx.de (V.W.)

**Keywords:** non-prescription drugs, community pharmacies, consultation, patient simulation, emergency contraception, ulipristal acetate, Germany

## Abstract

Background: In Germany, there are two different active substances, levonorgestrel (LNG) and ulipristal acetate (UPA), available as emergency contraception (the “morning after pill”) with UPA still effective even 72 to 120 h after unprotected sexual intercourse, unlike LNG. Emergency contraceptive pills have been available without a medical prescription since March 2015 but are still only dispensed by community pharmacies. The aim of this study was to determine the counselling and dispensing behaviour of pharmacy staff and the factors that may influence this behaviour in a scenario that intends that only the emergency contraceptive pill containing the active substance UPA is dispensed (appropriate outcome). Methods: A cross-sectional study was carried out in the form of a covert simulated patient study in a random sample of community pharmacies stratified by location in the German state of Mecklenburg-Vorpommern and reported in accordance with the STROBE statement. Each pharmacy was visited once at random by one of four trained test buyers. They simulated a product-based request for an emergency contraceptive pill, stating contraceptive failure 3.5 days prior as the reason. The test scenario and the evaluation forms are based on the recommended actions, including the checklist from the Federal Chamber of Pharmacies. Results: All 199 planned pharmacy visits were carried out. The appropriate outcome (dispensing of UPA) was achieved in 78.9% of the test purchases (157/199). A significant correlation was identified between the use of the counselling room and the use of a checklist (*p* < 0.001). The use of a checklist led to a significantly higher questioning score (*p* < 0.001). In a multivariate binary logistic regression analysis, a higher questioning score (adjusted odds ratio [AOR] = 1.41; 95% CI = 1.22–1.63; *p* < 0.001) and a time between 12:01 and 4:00 p.m. (AOR = 2.54; 95% CI = 1.13–5.73; *p* = 0.024) compared to 8:00 to 12:00 a.m. were significantly associated with achieving the appropriate outcome. Conclusions: In a little over one-fifth of all test purchases, the required dispensing of UPA did not occur. The use of a counselling room and a checklist, the use of a checklist and the questioning score as well as the questioning score and achieving the appropriate outcome are all significantly correlated. A target regulation for the use of a counselling room, an explicit guideline recommendation about the use of a checklist, an obligation for keeping UPA in stock and appropriate mandatory continuing education programmes should be considered.

## 1. Introduction

Sexual intercourse without contraception, an improperly used or broken condom, improperly applied regular contraception or sexual assaults can lead to unintended pregnancies [1]. The consequences of unintended pregnancies or the need to terminate a pregnancy affect not only the physical but also the social and emotional health of the women concerned and often do so over the long term [2]. It is estimated that in Germany about 34% [3] of all pregnancies are unintended (globally, about 44% [4]), of which about 43% [3] (globally, about 56% [4]) end in termination of a pregnancy. Emergency contraception (EC) plays an important role in preventing this situation. EC can be divided into copper intrauterine devices (Cu-IUD) and oral hormonal methods [5].

The Cu-IUD is the most reliable EC method with a failure rate of <0.1%, and it is also effective after ovulation has occurred [6]. However, access to this form of EC is made difficult in that a gynaecologist must insert the Cu-IUD, meaning that this is not the preferred method for many users [5]. In Germany, it is only used in isolated cases [7]. Of the oral hormonal methods, the Yuzpe method (oestrogen–progesterone combination comprising two doses of 50 µg ethinylestradiol and 0.25 mg levonorgestrel) is also no longer recommended in Germany for post-coital contraception [7] because it is less effective than other EC methods and is also associated with more adverse events [1,6,7]. Because mifepristone is not available in Germany for this indication [7,8]—unlike some other countries [5,9]—the oral hormonal methods are concentrated on emergency contraceptive pills (or “morning after pills”) containing the active substances levonorgestrel (LNG) and ulipristal acetate (UPA).

LNG is a synthetic gestagen that reduces the surge in luteinising hormone (LH), thus delaying ovulation. However, it must be taken before the surge in LH. UPA is a selective progesterone receptor modulator that also inhibits the LH surge but, unlike LNG, is still effective during the LH surge up to the LH peak [5,6,10]. Therefore, UPA is effective for up to 120 h after unprotected sexual intercourse or contraceptive failure. In contrast to UPA, LNG is effective up to 72 h after unprotected sexual intercourse or contraceptive failure, even though a moderate efficacy of up to 120 h is discussed in the international literature [5,6]. The superior effectiveness of UPA compared to LNG is also apparent in the first 24 to 72 h [11,12,13]. There are differences in terms of the statistical significance, however [11,12,13], which in Germany has led to different recommendations from gynaecology associations (generally UPA is considered superior) and the Federal Chamber of Pharmacies (BAK) (less than 72 h LNG or UPA) [14,15]. Both substances are more effective the sooner they are taken after unprotected sexual intercourse [7], which is why prompt access to the medication is of great importance. Analogous to many other countries [16], it has therefore been possible to obtain such preparations in Germany without a prescription since March 2015 [17], but the dispensing of these medications is still restricted to community pharmacies. Gynaecology associations and the German Medical Association (BÄK) argued against no-prescription dispensing of emergency contraceptive pills, citing the need for a medical consultation [18,19]. Not least, given this background [20], the BAK published corresponding recommendations including a checklist for quality assurance of the counselling provided when dispensing emergency contraceptive pills, whereby the use of this checklist is not explicitly advised in the recommendations [15]. The recommendations, which were developed with input from many experts across a range of organisations in various disciplines (e.g., BÄK, professional organisations and associations of gynaecologists, pharmaceutical OTC industry, government-controlled, private and church-based organisations and centres providing advice on sex education and family planning) [20], include the prerequisites for dispensing emergency contraception containing the active substances LNG or UPA for self-medication. Information about counselling and dispensing are also provided as well as the criteria regarding the limits of self-medication and referral to a doctor. Not only pharmacists but pharmacy technicians and pharmaceutical technical assistants are authorised to provide counselling for dispensing medicinal products such as the emergency contraceptive pill in Germany. Considering that the issue is still a sensitive one for many consumers, maintaining the privacy of the customer for this indication is a particularly important counselling criterion [21,22,23,24]. Nevertheless, both the Ordinance on the Operation of Pharmacies in Germany and the BAK recommendations expect that the counselling is conducted in private to maintain confidentiality and prevent other customers overhearing the counselling as far as possible [15,25]. There is as yet no legal obligation or a recommendation based on BAK guidelines to keep a separate counselling room, however [15,25]. 

The BAK recommendations stipulate the dispensing of only UPA 72 to 120 h after unprotected sexual intercourse or contraceptive failure, but it is not currently known to what extent pharmacies comply with these recommendations. The three studies available to date in Germany on the counselling and dispensing behaviour for emergency contraception have also not investigated this issue [26,27,28]. The study results are also based on self-assessment by pharmacies and may therefore be distorted, for example, due to social desirability [29,30]. Applying the simulated patient method (SPM), which is considered to be the “gold standard” [31] even if the relatively high administrative and financial costs and the comparably smaller sample sizes [31] as well as any intra- and inter-observer variabilities [32] are taken into account, can avoid such weaknesses because a realistic (counselling) situation can be simulated [33]. Unlike the many studies conducted in other countries [21,23,30,34,35,36,37,38,39,40,41,42,43,44,45], in Germany, there have not been any investigations to date on the counselling and dispensing of an emergency contraceptive pill using an SPM approach because the few scientific SP studies available for Germany on the counselling quality in pharmacies are always based on indications other than emergency contraception. Clear deficits in the quality of counselling have been identified in these studies [46,47,48,49,50,51]. 

The aim of this study was to identify the counselling and dispensing behaviour of pharmacy staff and the factors that may influence this behaviour in a scenario that intends that the emergency contraceptive pill containing only the active substance UPA is dispensed (appropriate outcome).

## 2. Methods

### 2.1. Design

A cross-sectional study design was chosen in accordance with the “STROBE Statement—Checklist of items that should be included in reports of cross-sectional studies” [52]. The quality of the counselling provided in the pharmacies visited is determined using the SPM, which has often been applied in international studies in the community pharmacy (CP) setting [33,53,54]. The SPM is a covert participatory observation [55] by a person, who in an ideal case cannot be differentiated from a real customer (simulated patient, SP), who visits a CP to simulate a real-life counselling situation based on a previously defined scenario. The data are then collected according to previously defined criteria using an assessment form and the CP is provided with performance feedback, if applicable [33]. 

### 2.2. Setting and Participation 

The test purchases took place between 1 July and 30 September, 2019 in the German state of Mecklenburg-Vorpommern (31 December, 2019: approx. 1.61 million residents, 23,216 km^2^ area, low population density of 69.3 residents/km^2^ [56]). Upon request by telephone, the Pharmacy Association of Mecklenburg-Vorpommern did not want to provide a list of all registered community pharmacies in the state. Consequently, the list of these pharmacies was identified using the pharmacy finder of the website Apotheken-Umschau.de [57]. All community pharmacies that had a postcode in the state of Mecklenburg-Vorpommern on the reference date of 1 June, 2019 using the postcode search of the pharmacy finder were included in the study. These hits were validated with a corresponding Google search. As a result, a basic population of N = 395 pharmacies was formed. A comparison with the most recently available information from the German Federal Chamber of Pharmacies (ABDA; Bundesvereinigung Deutscher Apothekerverbände) for the end of 2018 regarding the total number of pharmacies in Mecklenburg-Vorpommern [58] showed a 98% agreement. 

In Germany there have been no studies conducted to date on the dispensing of emergency contraceptive pills for a scenario in which dispensing UPA is mandatory. The degree of variability is therefore unknown. The minimum necessary sample size (n) was determined for a population size (N) of 395 and an error margin (e) of 0.05 using the following formula, which is based on a degree of variability of *p* = 0.5 and a 95% confidence interval [59]:(1)$$n=\;\frac{N}{1+N{\left(e\right)}^{2}}=\frac{395}{1+395{\left(0.05\right)}^{2}}=\frac{395}{1.9875}=198.74\;Population\;size=N\;|\;Margin\;of\;error=e$$

The assumed degree of variability of *p* = 0.5 maximises the required sample size. The 395 CPs were stratified by location to indicate whether they are urban or rural. Using the MS Excel random number generator, they were each then assigned a random number and the 199 participating CPs were then selected by simple random sampling from each stratum.

### 2.3. Scenario and Assessment

The recommendations, including the checklist published by the BAK, formed the basis of the test scenario that was developed (see Table 1) and the evaluation form used (see Table 2). The request for the “morning after pill” because of a broken condom that was simulated in the scenario is realistic because this reason was cited most frequently in requests for emergency contraception in pharmacies in a recent German study [26]. Dispensing the emergency contraception ellaOne^®^, the only preparation with the active substance UPA available on the German market [60], was defined as the appropriate outcome. According to the recommendation from the BAK, unprotected sexual intercourse 3.5 days ago must be considered the criterion for dispensing UPA instead of LNG [15]. 

The evaluation form includes 14 items, whereby the first nine evaluate whether appropriate questions were asked analogous to the BAK checklist. Based on the form, the decision was subsequently made by pharmacy staff whether the emergency contraceptive pill ellaOne^®^ with the active substance UPA is dispensed (item 10). It was also recorded whether LNG was erroneously dispensed instead of UPA (item 11). For the situation in which a medicinal product was dispensed, it was also evaluated according to the BAK recommendations whether the test buyers were informed about common adverse events such as headaches, nausea, dizziness, abdominal pain and cramps, dysmenorrhoea, vomiting, tiredness or breast tenderness [15] (item 12). Finally, the overall conditions of the counselling were also recorded, that is, whether the counselling was conducted in a separate room (item 13) and whether a checklist was visibly used to ensure the quality of the counselling (item 14).

For the evaluation form, only objective items were used to avoid a subjective assessment and thus latitude in the evaluation by the SPs (for example, on the friendliness of the pharmacy staff). Therefore, to complete the individual items, only dichotomous scales were used (closed yes/no questions).

### 2.4. Data Collection

Four female Master’s students from the Department of Health, Nursing, Management of the Neubrandenburg University of Applied Sciences acted as test buyers. They were selected on the basis of their participation in a 3-semester student research project. The students are all in the age group of 18 to 29 years, which corresponds to the age group of the primary users of emergency contraception [61]. Each pharmacy was visited once (a total of 199 test purchases), whereby the pharmacies were distributed randomly across the four SPs. 

Before the data collection was started, the test buyers underwent comprehensive training. Each test buyer first familiarised herself with the theoretical principles of the SP method, the test scenario itself and the contents of the evaluation form. The test buyers then each carried out four pre-tests for the scenario to confirm the functionality of the evaluation form and the test scenario and to train in the practical application of the SP method. The total of 16 pre-tests were carried out in pharmacies in Mecklenburg-Vorpommern that were not included in the stratified random sample. The pre-tests indicated that no changes to the test scenario or the evaluation form were required (see Figure 1).

The test purchases were carried out on different days of the week and at different times of the day. The SPs made their request to the pharmacy staff who first approached them. The SPs only provided additional information if they were then asked by the pharmacy staff to ensure that the information provided is consistent.

Along with the items on the evaluation form, the SPs planned, analogous to the international literature (see Table 3), to also collect a number of variables during and after the test purchases that may possibly affect the appropriate outcome. Unlike other international SP studies, the variable “pharmacy type” (chain vs. independent) was not recorded [35,37,44] because in Germany a pharmacy may only be operated by one pharmacist as the owner and in addition ownership is restricted to 1 primary pharmacy with up to 3 subsidiary pharmacies in the immediate neighbourhood [62].

So that, on the one hand, the most realistic counselling situation possible could be simulated and to identify any planned dispensing of medicinal products while at the same time avoiding unnecessary purchasing of medicinal products and thus waste (because the test buyers do not actually need the emergency contraceptive pill), in the cases in which a medicinal product was going to be dispensed, the test buyers informed the pharmacy staff just before making payment that they had left their wallet at home. 

In the literature, audio recordings are recommended for quality assurance of the test purchases [70]. For data privacy reasons, however, this was omitted because otherwise the pharmacies must be informed beforehand about the audio recordings and thus the study design is no longer covert. However, the corresponding evaluation form was completed by the test buyers immediately after visiting the pharmacy so that any distortions in the study results due to faulty recall by the test buyers could be minimised.

After evaluating the data collection, each pharmacy received a written performance feedback specific for each pharmacy including graphically edited benchmarking, whereby for each pharmacy any improvement or deterioration regarding the individual criteria was shown compared to the other anonymised pharmacies in the stratified random sample. In this way, the pharmacies were informed about the position of their competitors, so that ideally appropriate optimisation processes by the pharmacies investigated could be initiated based on the feedback with the aim of sustainably improving the quality of counselling provided.

### 2.5. Ethical Approval

According to the “Guideline for the use of mystery research in market and social research” [71], the data collected were anonymised and recorded in such a way that the pharmacies or the personnel involved could not be identified. To avoid a possible Hawthorne effect [72] and also a possible selection bias [73], the test purchases were carried out covertly—that is, without informing the pharmacies in advance—analogous to some other national [48,49,50,51] and international studies [21,33,38] and therefore pharmacies were not asked for consent to participate in advance. The lack of informed consent in advance was—analogous to the international literature [40,74,75]—resolved in that all pharmacies were informed about the procedure and the background of the study upon completion of the study. Recruited students provided their written informed consent to act as SPs. The study protocol was approved by the institutional ethics committee of the Neubrandenburg University of Applied Sciences (registration number: HSNB/KHM/152/19). 

### 2.6. Data Analysis

The data were entered using the four-eyes principle and analysed with SPSS Version 25 for Windows (IBM, Armonk, NY, USA). As part of the descriptive statistics, frequencies and percentages were determined. 95% confidence intervals for categorical data using bootstrapping were also reported. A Pearson’s chi-square test was performed to determine if interactions involving “counselling room used” were more likely to result in “visible use of a checklist”. Cramer’s V was reported as a measure of effect size. Applying the Kolmogorov–Smirnov test as well as the Shapiro–Wilk test indicated that the data did not have a normal distribution. Therefore, the median, interquartile range (IQR), minimum and maximum were calculated for continuous variables. With the help of the non-parametric Mann–Whitney U test, it was analysed whether “visible use of a checklist” led to a significantly higher median questioning score. Pearson’s r was reported as a measure of the effect size. A binomial logistic regression model was also developed to identify the influence of various independent variables (see Table 3) on the appropriate outcome (dispensing of the emergency contraceptive pill with the active substance UPA). All independent variables were checked for outliers and multicollinearity. Variables with a *p*-value less than 0.05 in the univariate analysis were included in the multivariate analysis. Odds ratios (OR), 95% confidence intervals, *p* values and as a measure of the effect size Cohen’s f^2^ were reported. A *p* value of less than 0.05 was considered to be significant in all analyses. 

## 3. Results

All 199 planned test purchases could be carried out (visit completion rate: 100%). Due to the exit strategy developed that resulted in the ending of the purchase, there were no direct costs. The total of 5080 km between the individual test purchases were driven in private vehicles and led to indirect costs totalling €1150 that were financed from the primary author’s own resources. Socio-demographic data for the pharmacies or the advising pharmacy staff are shown in Table 4. Most of the pharmacies tested were in local competition with one another and often did not have a quality certificate (62.8%, 125/199). The advising pharmacy staff were in most cases female (87.4%, 174/199) and aged between 30 and 49 years (57.3%, 114/199). In the cases in which it could be determined, the proportion of pharmacists (37.7%, 75/199) compared to non-pharmacists (pharmacy technicians and pharmaceutical technical assistants) (40.7%, 81/199) was almost equal. 

The appropriate outcome—dispensing of UPA—was achieved in 78.9% of the test purchases (157/199) (see Table 5). In 3.0% of the test purchases LNG (6/199) was dispensed while in 18.1% (36/199) no preparation was dispensed. There was a median questioning score of 5.0 (IQR 2.0–9.0) with a minimum score of 0 in 2.5% (5/199) of test purchases and a maximum score of 9 in 27.6% (55/199) of test purchases. The question about the time since the unprotected sexual intercourse took place was asked most frequently (93.5%, 186/199). Regarding the test purchases in which a medication was dispensed, information was provided about possible side effects by the pharmacy staff in 59.5% (97/163) of test purchases. 

Counselling was provided in a counselling room in slightly less than half of all the test purchases (44.2%, 88/199). In just over half of all test purchases, a checklist was visibly used for the counselling (53.8%, 107/199). A significant correlation could be found between the use of a counselling room and the use of a checklist (Pearson’s chi-square test; χ2(1) = 104.355, *p* < 0.001, V = 0.724) whereby the effect size V according to Cohen [76] corresponded to a ‘large’ effect. In 43.7% (87/199) of the test purchases, the counselling was not carried out in a counselling room nor was a checklist used. In contrast to this, in 41.7% (83/199) of the test purchases, a counselling room was sought out, and a checklist was used for the counselling. 

In addition, the use of a checklist led to a significantly higher questioning score (Mann–Whitney U test; U = 385.500, *p* < 0.001, r = 0.806), whereby the effect size r according to Cohen [76] corresponded to a ‘large’ effect. If a checklist was visibly used, this led to a median questioning score of 9.0 (IQR 7.0–9.0) with a minimum score of 2 in 0.9% (1/107) and a maximum score of 9 in 51.4% (55/107) of test purchases. If a checklist was not visibly used, the median questioning score was 2.0 (IQR 1.0–3.0) with a minimum score of 0 in 5.4% (5/92) and a maximum score of 7 in 2.2% (2/92) of test purchases. 

Table 6 shows the binary logistic regression model. As part of the bivariate analysis, three (age of the pharmacy staff, time of the test purchase, questioning score) of nine predictor variables had a *p*-value of <0.05 and were included in the multivariate logistic regression model. A time between 12:01 and 4:00 p.m. (AOR = 2.54; 95% CI = 1.13–5.73; *p* = 0.024) compared to 8:00 to 12:00 a.m. and a higher questioning score (AOR = 1.41; 95% CI = 1.22–1.63; *p* < 0.001) were significantly associated with dispensing of UPA (appropriate outcome). The location of the pharmacy, the presence of a quality certificate, a queue and the age, gender and professional group of the pharmacy staff as well as the SP number did not have any significant effect on the appropriate outcome. The model yielded a Nagelkerke R^2^ value of 0.255, which corresponds to a Cohen’s f^2^ of 0.342 and thus to a ‘medium’ effect size [76]. 

## 4. Discussion

The appropriate outcome, that is, the dispensing of UPA, was achieved in almost 80% of all test purchases. International SP studies on the quality of counselling provided for “morning after pills” in the Democratic Republic of Congo, Kenya and India show a similarly high case resolution of 74% [38], 82% [34] and 86% [30], while in Australia and Switzerland values of 95% [21] and even 100% [23] were achieved. In contrast, in one Australian SP study, this value turned out to be considerably lower with 24% [35]. The highly divergent values in some cases must be viewed and interpreted in light of the very different scenarios used. 

The fact that in slightly more than 20% of all test purchases, a medicinal product was not dispensed, or the wrong medicinal product was dispensed is problematic [77]. This situation is all the more significant for such an indication in which the consequence of providing the wrong advice or not dispensing the correct medicinal product—possibly an unwanted pregnancy—must be considered to be very high. 

Possible reasons for not dispensing UPA that were also seen in the international SP studies of oral emergency contraceptive pills is the unavailability of the preparation UPA [78,79] and a lack of knowledge on behalf of the pharmacy staff [80]. Appropriate mandatory continuing education programs could reduce the issue of a lack of knowledge [81]. The unavailability of the UPA preparation could be minimised by mandating keeping the preparation in stock, especially as UPA is the only recommended active substance on the market for a specific time window and also because it should be taken as soon as possible.

In contrast to a Saudi Arabian study [64], the time proved to be a significant influencing factor on the appropriate outcome even though the type of outcome between the two studies is not comparable (UPA dispensed vs. antibiotics dispensed without a prescription). This creates a need for further research about the possible reasons, especially as the best counselling was provided not in the morning as expected on the basis of the model of the performance curve [82] but instead from midday to the late afternoon.

Analogous to the international literature [67,83] the questioning score had a significant effect on the appropriate outcome. Regarding the individual questions, the question about when the unprotected sexual intercourse took place proved to be the “master question” because only then could the correct decision—dispensing UPA—be made at all. The SP studies conducted in Switzerland, Scotland and Australia showed similarly high values to those in this investigation with values of 100% [23,45], 93% [35] and 88% [36] respectively. In contrast, in one SP study in the Democratic Republic of Congo, this question was only asked in 7% of the test purchases [38], and in a Turkish SP study it was asked in only one of 155 test purchases [41]. In an Indian and a Brazilian SP study this question was not asked once in 70 [30] and 122 [44] test purchases respectively.

A significant correlation between the questioning score and the visible use of a checklist was apparent, analogous to the international literature [23,35]. However, a checklist was only used in a little over half of all test purchases. In contrast, in a qualitative German study, it was reported that almost all pharmacies stated that they used a checklist [27]. Such discrepancies—worse results in SP studies than in self-reported studies of the same issue—are also seen in the international mixed-methods literature for requests for medicinal products other than oral emergency contraceptive pills [84,85]. Internationally, a checklist was only used in 10% of all test purchases in an Australian SP study [21] whereas this value increased to 83% in another Australian SP study that was conducted because a checklist had since been developed and recommended by the Pharmaceutical Society of Australia [35]. However, in a more recent Australian SP study, a checklist was not used in a single test purchase [36]. The use of a checklist is also recommended in the guidelines for Switzerland, which led to an application rate of 99% in a recent Swiss SP study [23]. In contrast to an Australian SP study [35], our study showed not only a significant correlation between ”checklist” and “questioning score” but the questioning score also has a significant effect on the dispensing of UPA. It, therefore, appears to be advisable—despite criticism from pharmacy staff in a qualitative German study that the BAK checklist, in particular, is overly detailed and thus requires considerable time [27]—to introduce an explicit guideline recommendation about the use of a checklist in Germany as well. However, optimising or shortening the checklist should be considered to increase its acceptance.

Although there is a significant correlation between the use of a checklist and the use of a counselling room, in this study, a counselling room was only used in a little less than half of the test purchases. In contrast, two-thirds of the interviewed pharmacy staff in the qualitative German study indicated using a counselling room [27]. Internationally, there is an inconsistent picture regarding the use of a counselling room. For example, in an Australian SP study, only 9% of the test purchases were conducted in a counselling room [36] while in an Indian SP study this value was as high as 47% [30]. In contrast to this, an extra counselling room or a quiet area in the pharmacy was used in 90% of the consultations in another Australian SP study [21]. In Switzerland, pharmacists are supposed to carry out the consultation for the “morning after pill” in a separate counselling room, which was again implemented in 94% of test purchases in a corresponding SP study—analogous to the recommended use of a checklist [23]. To protect the privacy of patients for this sensitive issue, such a target regulation should therefore be considered for Germany as well. Furthermore, the flow of information between the consumer and the pharmacy staff could also be improved as a result, which was established in a qualitative Australian study [22]. Due to the correlation between the use of a counselling room and the use of a checklist, it is also likely that the use of a checklist could be increased in this way [21]. 

There is a need for further research in Germany regarding access to and the quality of counselling provided for emergency contraceptive pills for minors [30], male consumers [39] and victims of sexual assault [43]. It should also be analysed in future whether pharmacy staff provide additional important information—about sexually transmissible infections, for example—during the counselling [35]. Future studies should also analyse the size of any differences between SP results and self-reported results by pharmacy staff by using a mixed-methods approach regarding the counselling provided for oral emergency contraception.

### Strengths and Limitations

This study is the first in Germany that investigated the counselling and dispensing behaviour of pharmacies for the “morning after pill” using an SP method. The SP study design used is particularly advantageous because unlike other data collection methods, both the effect of social desirability and the Hawthorne effect can be avoided. The results—at least for one German state—can also be generalised because of the stratified random sampling used. There is also no selection bias because there is no option for opting out. The study could also be fully implemented as planned due to the 100% visit completion rate. 

Because the study results only refer to one German state, future studies should be expanded to additional states or all of Germany. However, this would be an ambitious undertaking in light of the relatively high data collection costs. Furthermore, only one specific counselling and dispensing scenario (mandatory UPA dispensing) for oral emergency contraceptive pills were used for the data collection. Other scenarios could therefore lead to different results [48,67]. Because only students in a certain age range were used as SPs, it also cannot be ruled out that the counselling and dispensing behaviour would differ for other age groups (such as adolescents) or women with different levels of education (such as customers with secondary school education only). The study also did not differentiate between non-dispensing of UPA due to a lack of availability and non-dispensing due to incorrect counselling. Not dispensing UPA due to incorrect counselling would be classified as particularly serious because the test buyers would not receive UPA as a result, even though it would be the suitable active substance. However, not dispensing UPA due to a lack of availability would be problematic because the medication is supposed to be taken as soon as possible. Future studies should, therefore, record this differentiation because it would provide important information about possible optimisation measures. Furthermore, this is a cross-sectional study, and therefore, no causal relationship can be created between the variables studied. Because the study design (no audiotaping and no second observer) meant that the test buyers had to remember the recorded items until they left the pharmacy and many items had to be recorded (14 items for counselling as well as several influencing factor items), collecting additional important items (such as information about taking the medication as soon as possible or about taking the medication again in case of vomiting) was also omitted to prevent any recall bias. Immediate feedback after the test purchase would also have been desirable because then the pharmacy staff’s memory of a specific counselling situation would have still been fresh. However, there is a risk that the pharmacy staff inform their colleagues in the neighbourhood about the test purchases. As a result, it was also not possible to ask the counselling pharmacy staff about their age, meaning that there may be certain distortions in the estimations of the age variable.

## 5. Conclusions

In almost four-fifths of all test purchases, the appropriate outcome was achieved. This meant, however, that in a little over one-fifth of all test purchases, the required dispensing of UPA did not occur, which could have led to serious consequences such as an unwanted pregnancy. The use of a counselling room and a checklist, the use of a checklist and the questioning score as well as the questioning score and achieving the appropriate outcome are all significantly correlated. A target regulation for the use of a counselling room, an explicit guideline recommendation about the use of a checklist, an obligation for keeping UPA in stock and appropriate mandatory continuing education programs should be considered. Surprisingly, the time of day proved to be a significant influencing factor on the appropriate outcome, which requires further investigation.

## Figures and Tables

**Figure 1 ijerph-17-06720-f001:**
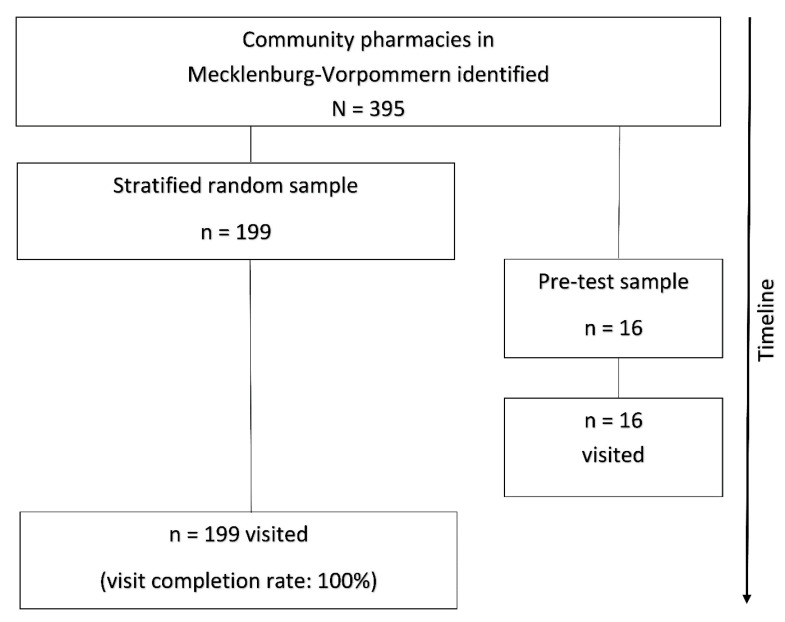
The design and flow of the study.

**Table 1 ijerph-17-06720-t001:** Test scenario.

Scenario
The test buyers enter the pharmacy and ask for oral emergency contraception without having a specific product in mind (product-based query).
When questioned by the pharmacy staff, the following information is provided: - Real age of the test buyer - “Morning after pill” is needed because of a broken condom - Unprotected sexual intercourse was 3.5 days ago - Last period was 11 days ago - An existing pregnancy is not suspected - No nausea with urge to vomit, no vomiting, no medical conditions - Not breastfeeding - Not taking other medications - Not repeated application

**Table 2 ijerph-17-06720-t002:** Evaluation form.

Items	Yes	No
1. Age of the patient?	1	0
2. Why was the “morning after pill” requested?	1	0
3. When did the unprotected sexual intercourse take place?	1	0
4. When was the customer’s last period?	1	0
5. Are there indications of an existing pregnancy?	1	0
6. Are there existing health problems or are nausea with an urge to vomit or vomiting present?	1	0
7. Is the customer breastfeeding?	1	0
8. Current or regular use of other medications?	1	0
9. Is this a repeated application?	1	0
10. Dispensing the emergency contraceptive pill (UPA) (‘appropriate outcome’)	1	0
11. Dispensing the emergency contraceptive pill (LNG)	1	0
12. Explanation of side effects	1	0
13. Counselling room used	1	0
14. Visible use of a checklist	1	0

**Table 3 ijerph-17-06720-t003:** Possible influencing factors as well as time and type of data collection.

Possible Influencing Factors [Literature Source *]	Time of Data Collection	Type of Data Collection
Location of the pharmacy [47] as an indicator for urban/rural	Before the test purchase because stratification variable	Precise measurement by allocating the number of pharmacies identified in the particular area
SP number [63]	During the test purchase	Exact measurement by assigning a number to each SP
Age of the pharmacy staff [64]	During the test purchase	Estimate based on visual impression by SP
Gender of the pharmacy staff [64]	During the test purchase	Exact measurement using visual impression of the SP
Queue—patients waiting after the SP [65]	During the test purchase	Exact measurement using visual impression of the SP
Time of the test purchase [66]	During the test purchase	Exact measurement using the SP’s watch
Professional group of the pharmacy staff [67]	During and after the test purchase	Exact measurement based on the name tag and, if necessary, using a telephone query by the SP after completing all the test purchases
Pharmacy quality certificate [68]	After the test purchase	Precise measurement using a telephone query by the SP after completing all the test purchases
Questioning score [69]	After the test purchase	Precise measurement by summing the dichotomous evaluation of the nine individual questions (minimum possible score of 0 points and maximum possible score of 9 points)

Note: * The possible influencing factors were taken from the specific literature sources.

**Table 4 ijerph-17-06720-t004:** Socio-demographic data for the pharmacies or the advising pharmacy staff.

	Frequency (*n*)	Percentage (%)
All pharmacies	199	100
Location of the pharmacy		
1 pharmacy in the area	37	18.6
2–4 pharmacies in the area	58	29.1
5–19 pharmacies in the area	46	23.1
≥20 pharmacies in the area	58	29.2
Pharmacy quality certificate		
No	125	62.8
Yes	51	25.6
Not able to be determined	23	11.6
Age of the pharmacy staff		
<30	27	13.6
30–49	114	57.3
≥50	58	29.1
Gender of the pharmacy staff		
Male	25	12.6
Female	174	87.4
Professional group of the pharmacy staff		
Pharmacist	75	37.7
Non-pharmacist	81	40.7
Not able to be determined	43	21.6

**Table 5 ijerph-17-06720-t005:** Assessment items (*n* = 199).

	Yes		
	Frequency (*n*)	Percentage (%)	95% CI
1. Age of the patient?	118	59.3	52.3–66.3
2. Why was the “morning after pill” requested?	103	51.8	45.2–58.8
3. When did the unprotected sexual intercourse take place?	186	93.5	89.9–96.5
4. When was the customer’s last period?	133	66.8	59.8–73.4
5. Are there indications of an existing pregnancy?	92	46.2	39.2–53.3
6. Are there existing health problems or are nausea with an urge to vomit or vomiting present?	94	47.2	39.7–54.3
7. Is the customer breastfeeding?	82	41.2	34.2–48.2
8. Current or regular use of other medications?	125	62.8	55.3–70.4
9. Is this a repeated application?	110	55.3	48.2–62.8
10. Dispensing the emergency contraceptive pill (UPA)(‘appropriate outcome’)	157	78.9	72.9–84.4
11. Dispensing the emergency contraceptive pill (LNG)	6	3.0	1.0–5.5
12. Explanation of side effects	97	59.5	52.0–67.1
13. Counselling room used	88	44.2	37.2–51.8
14. Visible use of a checklist	107	53.8	46.7–61.3

**Table 6 ijerph-17-06720-t006:** Possible influencing factors on the recommendation of UPA.

Possible Influencing Factors and Categories	*n* (%) Total 199 (100)	*n* (%) Recommendation 157 (78.9)	*n* (%) No Recommendation 42 (21.1)	COR (95% CI)	*p*-Value	AOR (95% CI)	*p*-Value
Location of the pharmacy							
1 pharmacy in the area	37 (100)	27 (73.0)	10 (27.0)	1			
2–4 pharmacies in the area	58 (100)	49 (84.5)	9 (15.5)	2.02 (0.73–5.57)	0.176		
5–19 pharmacies in the area	46 (100)	37 (80.4)	9 (19.6)	1.52 (0.55–4.26)	0.423		
≥ 20 pharmacies in the area	58 (100)	44 (75.9)	14 (24.1)	1.16 (0.45–2.99)	0.752		
Pharmacy quality certificate							
No	125 (100)	102 (81.6)	23 (18.4)	1			
Yes	51 (100)	40 (78.4)	11 (21.6)	0.82 (0.37–1.84)	0.629		
Not able to be determined	23 (100)	15 (65.2)	8 (34.8)	0.42 (0.16–1.12)	0.082		
Age of the pharmacy staff							
<30	27 (100)	17 (63.0)	10 (37.0)	1		1	
30–49	114 (100)	91 (79.8)	23 (20.2)	2.33 (0.94–5.75)	0.067	2.05 (0.74–5.64)	0.166
≥50	58 (100)	49 (84.5)	9 (15.5)	3.20 (1.11–9.21)	0.031	2.68 (0.84–8.54)	0.095
Gender of the pharmacy staff							
Male	25 (100)	18 (72.0)	7 (28.0)	1			
Female	174 (100)	139 (79.9)	35 (20.1)	1.54 (0.60–3.99)	0.369		
Professional group of the pharmacy staff							
Pharmacist	75 (100)	63 (84.0)	12 (16.0)	1			
Non-pharmacist	81 (100)	63 (77.8)	18 (22.2)	0.67 (0.30–1.50)	0.326		
Not able to be determined	43 (100)	31 (72.1)	12 (27.9)	0.49 (0.20–1.22)	0.126		
Time of the test purchase							
8:00 a.m.–12:00 p.m.	85 (100)	61 (71.8)	24 (28.2)	1		1	
12:01 p.m.–4:00 p.m.	92 (100)	78 (84.8)	14 (15.2)	2.19 (1.05–4.59)	0.037	2.54 (1.13–5.73)	0.024 *
4:01 p.m.–8:00 p.m.	22 (100)	18 (81.8)	4 (18.2)	1.77 (0.54–5.77)	0.343	1.51 (0.41–5.56)	0.533
Queue							
No	144 (100)	115 (79.9)	29 (20.1)	1			
Yes	55 (100)	42 (76.4)	13 (23.6)	0.82 (0.39–1.71)	0.589		
SP number							
1	46 (100)	37 (80.4)	9 (19.6)	1			
2	55 (100)	40 (72.7)	15 (27.3)	0.65 (0.25–1.66)	0.367		
3	49 (100)	42 (85.7)	7 (14.3)	1.46 (0.50–4.31)	0.493		
4	49 (100)	38 (77.6)	11 (22.4)	0.84 (0.31–2.26)	0.731		
Questioning score ^a^	5.0 (2.0–9.0)	7.0 (3.0–9.0)	2.5 (1.0–4.0)	1.39 (1.21–1.60)	<0.001	1.41 (1.22–1.63)	<0.001 *

^a^ Median (Interquartile range; IQR); Abbreviations: COR = Crude Odds Ratio; AOR = Adjusted Odds Ratio. * significant at *p* < 0.05.

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
