# Peer review of "The Quality of Counselling for Oral Emergency Contraceptive Pills—A Simulated Patient Study in German Community Pharmacies"

_ijerph, 2020, doi:10.3390/ijerph17186720_

Round 1
Reviewer 1 Report
Here they are my comments:
Abstract: Please add the practicel implications of your findings for policy making, education and management. What should be done based on your findings to improve the current condition?
Methods: Table 2, what 'Explanation of adverse events' means? Doo you mean medications' side effects or drug reactions?
Did you check that there were no differences between the SPs also with regarding to collecting data and reporting them? How?
A figure could help with summarising the study method.
Table 6 is very disorganised and difficuly to read and understand. That could be fine the table for the logestic regression analysis as the suppliment to this article.
Conclusion: To be more practical, please suggest the implications of your findings for policy making, etc. What should be done based on your findings to improve the current condition?
Author Response
We would like to thank the reviewer for taking the time to review our manuscript and also for the very helpful and detailed comments, recommendations and questions. See our point-by-point response to the reviewer‘s comments below. Reviewer 1:
Here they are my comments:
Abstract: Please add the practicel implications of your findings for policy making, education and management. What should be done based on your findings to improve the current condition?
Thank you for this comment. A further sentence has been added accordingly.
Methods: Table 2, what 'Explanation of adverse events' means? Doo you mean medications' side effects or drug reactions?
Thank you for this comment. We mean ‘side effects‘, therefore 'Explanation of adverse events‘ was replaced with 'Explanation of side effects‘ in Table 2 and Table 5.
Did you check that there were no differences between the SPs also with regarding to collecting data and reporting them? How?
Thank you for this comment. Therefore the influence of an independent variable „SP number“ on the appropriate outcome was checked (see the additional reference no. 62, table 3 and table 6). As a result, no significant differences were found between the four SPs.
A figure could help with summarising the study method.
Thank you for this comment. A figure has been added accordingly.
Table 6 is very disorganised and difficuly to read and understand. That could be fine the table for the logestic regression analysis as the suppliment to this article.
Thank you for this comment. You are absolutely right, we assume that this error (and some more ones, see below) occurred while converting the word document to a pdf document by the MDPI editorial system. Therefore we informed the MDPI editorial office about these problems directly after our first submission, See our email with the following errors:
- Table 6 must be in landscape format, otherwise the numbers are difficult to read,
- Please adjust the numbering of the references analogous to the word file,
- Please standardize the different references (sometimes thicker) in the reference list.
Conclusion: To be more practical, please suggest the implications of your findings for policy making, etc. What should be done based on your findings to improve the current condition?
Thank you for this comment. A further sentence has been added accordingly.
Reviewer 2 Report
Thank you for giving me the opportunity to review this work. In general the wording and structure of the article is adequate.Summary
Include in summary results when the OR values and their CI95 from the multivariate analysis were statistically significant.
Results
The authors must fix the formats of tables 5 and 6, because they are misconfigured. Especially table 6 is not very readable and should be horizontal. ORs and their ranges should only be written to two decimal places.
In Table 6 the percentages of recommendation and no recommendation should be given on the total of each category, not on the total of each variable. In this way it will be possible to visually better compare that it has a greater or lesser degree of correct recommendation.
References
Check the font size of some quotes 15-19
Author Response
We would like to thank the reviewer for taking the time to review our manuscript and also for the very helpful and detailed comments, recommendations and questions. See our point-by-point response to the reviewer‘s comments below.
Reviewer 2: Thank you for giving me the opportunity to review this work. In general the wording and structure of the article is adequate.
Thank you for this general comment.
Summary
Include in summary results when the OR values and their CI95 from the multivariate analysis were statistically significant.
Thank you for this comment. All statistically significant variables with OR values and their CI95 from the multivariate analysis have been included.
Results
The authors must fix the formats of tables 5 and 6, because they are misconfigured. Especially table 6 is not very readable and should be horizontal.
Thank you for this comment. You are absolutely right, we assume that this error (and some more ones, see below) occurred while converting the word document to a pdf document by the MDPI editorial system. Therefore we informed the MDPI editorial office about these problems directly after our first submission, See our email with the following errors:
- Table 6 must be in landscape format, otherwise the numbers are difficult to read,
- Please adjust the numbering of the references analogous to the word file,
- Please standardize the different references (sometimes thicker) in the reference list.
ORs and their ranges should only be written to two decimal places.
Thank you for this comment. ORs and their ranges have been modified accordingly.
In Table 6 the percentages of recommendation and no recommendation should be given on the total of each category, not on the total of each variable. In this way it will be possible to visually better compare that it has a greater or lesser degree of correct recommendation.
Thank you for this comment. Table 6 has been revised accordingly.
References
Check the font size of some quotes 15-19
Thank you for this comment. You are absolutely right, we assume that this error (and some more ones, see below) occurred while converting the word document to a pdf document by the MDPI editorial system. Therefore we informed the MDPI editorial office about these problems directly after our first submission, See our email with the following errors:
- Table 6 must be in landscape format, otherwise the numbers are difficult to read,
- Please adjust the numbering of the references analogous to the word file,
- Please standardize the different references (sometimes thicker) in the reference list.